# Memory T Cells in Respiratory Virus Infections: Protective Potential and Persistent Vulnerabilities

**DOI:** 10.3390/medsci13020048

**Published:** 2025-04-29

**Authors:** Henry Sutanto, Febrian Ramadhan Pradana, Galih Januar Adytia, Bagus Aditya Ansharullah, Alief Waitupu, Bramantono Bramantono, Deasy Fetarayani

**Affiliations:** 1Internal Medicine Study Program, Department of Internal Medicine, Faculty of Medicine, Universitas Airlangga, Surabaya 60132, Indonesia; henry1988md@gmail.com (H.S.); febrianrp16@gmail.com (F.R.P.); galihja@gmail.com (G.J.A.); aditya08bagus@gmail.com (B.A.A.); alief.waitupu@gmail.com (A.W.); 2Department of Internal Medicine, Dr. Soetomo General Academic Hospital, Surabaya 60286, Indonesia; 3Division of Tropical and Infectious Diseases, Department of Internal Medicine, Faculty of Medicine, Universitas Airlangga, Surabaya 60132, Indonesia; 4Division of Allergy and Clinical Immunology, Department of Internal Medicine, Faculty of Medicine, Universitas Airlangga, Surabaya 60132, Indonesia

**Keywords:** memory T cells, lymphocyte, adaptive immunity, viral infection, immunology

## Abstract

Respiratory virus infections, such as those caused by influenza viruses, respiratory syncytial virus (RSV), and coronaviruses, pose a significant global health burden. While the immune system’s adaptive components, including memory T cells, are critical for recognizing and combating these pathogens, recurrent infections and variable disease outcomes persist. Memory T cells are a key element of long-term immunity, capable of responding swiftly upon re-exposure to pathogens. They play diverse roles, including cross-reactivity to conserved viral epitopes and modulation of inflammatory responses. However, the protective efficacy of these cells is influenced by several factors, including viral evolution, host age, and immune system dynamics. This review explores the dichotomy of memory T cells in respiratory virus infections: their potential to confer robust protection and the limitations that allow for breakthrough infections. Understanding the underlying mechanisms governing the formation, maintenance, and functional deployment of memory T cells in respiratory mucosa is critical for improving immunological interventions. We highlight recent advances in vaccine strategies aimed at bolstering T cell-mediated immunity and discuss the challenges posed by viral immune evasion. Addressing these gaps in knowledge is pivotal for designing effective therapeutics and vaccines to mitigate the global burden of respiratory viruses.

## 1. Introduction

Respiratory viruses, including influenza, respiratory syncytial virus (RSV), and coronaviruses, are significant causes of morbidity and mortality worldwide. These pathogens infect the respiratory tract, leading to a range of diseases from mild upper respiratory tract infections to severe pneumonia. Influenza and RSV primarily target children and older adults, with annual outbreaks causing significant healthcare burdens [1]. Similarly, the recent severe acute respiratory syndrome coronavirus 2 (SARS-CoV-2) pandemic underscored the devastating impact of coronaviruses on global health [2,3]. These viruses often evade immunity and lead to recurrent infections, necessitating further understanding of host immune responses. Adaptive immunity, comprising humoral and cellular components, plays a pivotal role in combating respiratory viruses. Memory T cells, a subset of adaptive immune cells, are essential for long-term immunity and protection against reinfection. They are activated during primary infection and persist to provide a faster and more robust response upon subsequent exposure to the same pathogen. In the respiratory tract, tissue-resident memory T cells (TRM) are particularly crucial as they provide localized immune protection at the site of infection [4]. However, the effectiveness of memory T cells varies across different respiratory viruses, with some pathogens, inducing short-lived or suboptimal memory responses [5].

Despite the generation of immune memory, respiratory virus infections frequently recur. Several factors contribute to this phenomenon. Firstly, respiratory viruses exhibit significant genetic variability, allowing them to evade previously established immune responses [1]. Secondly, the quality and duration of T cell memory differ depending on the virus and the host’s immune status. For instance, RSV-specific memory T cells are often short-lived and ineffective in preventing reinfection [6]. Additionally, immune responses in the respiratory tract must balance virus clearance with minimizing tissue damage, leading to immunoregulatory mechanisms that may impair memory T cell function [7]. Understanding these challenges is critical for developing effective vaccines and therapeutics targeting respiratory viruses. The aim of this review is to critically examine the role of memory T cells in the immune response to respiratory virus infections, exploring their potential to provide protection and the challenges that limit their efficacy. By synthesizing current research, the review seeks to clarify why infections often recur despite the presence of immune memory, addressing factors such as viral immune evasion, anatomical barriers, and host-specific influences. Additionally, it aims to highlight emerging strategies to enhance memory T cell responses through innovative vaccine designs and therapeutic interventions, ultimately providing insights for reducing the burden of respiratory viruses globally.

## 2. Memory T Cells: Overview

### 2.1. Definition and Types of Memory T Cells

Memory T cells are a specialized subset of T lymphocytes within the adaptive immune system that persist after the resolution of an infection. These cells provide long-lasting immunity by mounting a faster and stronger immune response upon re-exposure to the same pathogen. Memory T cells are generated during the primary immune response and play a critical role in protecting against recurrent infections, especially in mucosal tissues like the respiratory tract. Their longevity and ability to recall antigens make them pivotal in immune defense and vaccine development [4]. Memory T cells are categorized into three main types based on their phenotypic markers, functional properties, and localization: central memory T cells (TCM), effector memory T cells (TEM), and TRM (Table 1). TCM cells are primarily found in secondary lymphoid organs such as lymph nodes and spleen. They express high levels of CCR7 and CD62L, which allow them to circulate through lymphoid tissues and maintain their proliferative capacity. TCM cells are relatively less differentiated and can proliferate robustly upon antigen re-exposure, subsequently differentiating into effector T cells to combat infections [8]. TEM cells are found in the blood and peripheral tissues, including the respiratory tract. They lack the expression of CCR7 and CD62L, which restricts their circulation in lymphoid organs. Instead, TEM cells are highly differentiated and poised for immediate effector functions such as cytokine production and cytotoxic activity upon encountering antigens. This rapid-response ability is particularly valuable in controlling acute respiratory viral infections [5]. TRM cells are localized in non-lymphoid tissues, including the lung mucosa, and do not recirculate. These cells are characterized by the expression of surface markers such as CD69 and CD103, which facilitate their retention in tissues. TRM cells are critical for providing localized immunity at the site of infection, making them particularly effective against respiratory viruses. Studies suggest that TRM cells in the lungs are superior to circulating memory T cells in controlling respiratory viral infections due to their proximity to the site of entry [4]. However, the maintenance of TRM cells and their effectiveness in different viral contexts remain areas of active research.

Memory T cell subsets are distinguished by specific surface markers that reflect their migration, function, and localization patterns (Table 2). TCM cells express CD45RO, CCR7, and CD62L, which guide them to lymph nodes and support long-term surveillance and proliferative capacity [9]. In contrast, TEM cells also express CD45RO but lack CCR7 and CD62L, allowing them to circulate in peripheral tissues where they can rapidly exert effector functions upon antigen re-encounter [10,11]. TRM cells, distinct in their permanent residence within non-lymphoid tissues, are typically identified by CD69 and CD103 expression, which prevent tissue egress and mediate epithelial adhesion, respectively [12]. CD49a is another TRM marker linked to cytotoxic potential and retention in tissues like the lung [13,14]. Additionally, markers such as CX3CR1 help distinguish cytotoxic TEM populations, while S1PR1 is downregulated in TRM to prevent exit from tissues [15]. Finally, CD127 (interleukin [IL]-7Rα) is broadly expressed in memory subsets, supporting survival and homeostasis, while PD-1, often elevated in TRM, indicates chronic stimulation in barrier tissues [16]. These markers not only define memory T cell identity, but are also central to their roles in long-term immunity and tissue-specific responses.

### 2.2. Mechanisms of Memory T Cell Generation During Viral Infections

Memory T cell generation is a complex process that occurs during the adaptive immune response to viral infections (Figure 1). It involves the activation, expansion, differentiation, and eventual persistence of antigen-specific T cells. This process ensures that the immune system retains the ability to respond more effectively to future encounters with the same pathogen. The generation of memory T cells begins with the activation of naïve T cells upon recognition of viral antigens presented by antigen-presenting cells (APCs), such as dendritic cells. During respiratory viral infections, dendritic cells in the lungs play a pivotal role in capturing and presenting viral antigens to T cells in the draining lymph nodes. This interaction triggers T cell activation, proliferation, and differentiation into effector T cells capable of eliminating the infection [17]. This priming event, influenced by the strength and duration of T cell receptor (TCR) signaling, co-stimulation, and cytokines such as IL-2 and IL-12, initiates clonal expansion and the generation of short-lived effector T cells (TECs) and memory precursors (MPECs). Following the resolution of infection, most effector T cells undergo apoptosis, while a subset transitions into memory T cells. This fate decision is regulated by transcriptional and epigenetic mechanisms, with transcription factors such as TCF1 and Bcl-6 promoting memory lineage commitment, while others like Blimp-1 and T-bet favor effector fates. Memory T cells continue to mature over time into distinct subsets, including TCM, TEM, and TRM cells, each defined by unique surface markers and migratory patterns. TCM cells retain lymphoid homing receptors like CCR7 and CD62L, while TEM and TRM cells express markers like CD45RO and CD69 but lack lymphoid-homing capacity, allowing them to surveil peripheral tissues [18]. Several models of memory T cells’ differentiation have been proposed in literatures [19,20,21]. The differentiation into memory T cells is influenced by several factors, including the strength and duration of antigenic stimulation, cytokine signaling, and the metabolic state of the T cells. For example, IL-7 and IL-15 are critical cytokines that support the survival and homeostatic proliferation of memory T cells, while persistent exposure to high antigen levels can impair the generation of high-quality memory cells [8]. The density of T cells within local tissues can also modulate the rate of memory differentiation through quorum sensing and cytokine sensitivity, particularly to IL-2 and IL-6 [22]. Additionally, memory formation may not always require repeated antigen stimulation. Single-cell lineage tracing has shown that a single antigen encounter can irreversibly commit naïve T cells to a differentiation trajectory culminating in memory formation.

The tissue microenvironment, particularly in the lungs, is crucial for shaping memory T cell subsets. TRM cells are established in the lung mucosa during respiratory viral infections. These cells are retained at the site of infection through interactions with local chemokines and adhesion molecules, such as CD69 and CD103, and are maintained independently of circulating memory T cells [4]. Additionally, low levels of persistent viral antigens presented by resident dendritic cells can help sustain memory T cell populations without causing overt inflammation [17]. Memory T cell formation is tightly linked to metabolic changes and epigenetic reprogramming. Effector T cells that transition into memory cells shift from a glycolytic metabolic state to one that relies on oxidative phosphorylation and fatty acid oxidation, which supports long-term survival and function. Concurrently, epigenetic modifications enable memory T cells to retain their transcriptional program, allowing for rapid activation and effector function upon re-exposure to the same pathogen [4]. These mechanisms underscore the complexity of memory T cell generation during viral infections and highlight their importance in long-term immunity.

### 2.3. Lifespan and Maintenance of Memory T Cells

Memory T cells are distinguished by their ability to persist long after the resolution of infection, providing long-lasting immunity. Their maintenance relies on complex cellular processes, homeostatic proliferation, cytokine signaling, and environmental cues. However, the lifespan and functionality of memory T cells vary across different subsets and infection contexts. Memory T cells exhibit a remarkable ability to survive for years or even decades in the absence of antigen re-stimulation. This longevity is particularly evident in TCM, which reside in secondary lymphoid organs and proliferate at low levels in response to homeostatic cytokines such as IL-7 and IL-15 [8]. TEM cells, which circulate in peripheral blood and tissues, typically have a shorter lifespan but are maintained through periodic reactivation by low-level antigen exposure or cytokine signals. TRM cells, which remain in non-lymphoid tissues such as the lungs, are specialized for localized immunity and depend on their microenvironment for longevity [4]. The maintenance of memory T cells is supported by several factors, such as cytokine signaling, metabolic adaptation, and tissue-specific factors. IL-7 and IL-15 are critical for the survival and homeostatic proliferation of memory T cells. These cytokines act by preventing apoptosis and maintaining metabolic activity in memory T cells, allowing them to persist in the absence of antigen [5]. In the case of TRM cells, local cytokines like transforming growth factor beta (TGF-β) and IL-33 play a role in maintaining their tissue residency and survival. Memory T cells shift their metabolism from glycolysis, used during effector phases, to oxidative phosphorylation and fatty acid oxidation. This metabolic state supports the long-term survival and readiness of memory cells to respond to reinfection [4]. For TRM cells, the local tissue environment is critical for their maintenance. Lung TRM, for example, rely on specific interactions with chemokines and adhesion molecules such as CD69 and CD103 to remain in the tissue. The absence of these signals can lead to their migration or loss of functionality [17]. Despite their durability, memory T cells can be affected by aging, persistent infections, and environmental factors. For example, chronic stimulation by residual antigens or repeated infections may lead to memory T cell exhaustion, reducing their protective efficacy [6]. Additionally, respiratory viruses such as RSV are known to generate memory T cells with limited lifespan and function, presenting unique challenges in immunity [5].

## 3. Protective Role of Memory T Cells Against Respiratory Viruses

Memory T cells play a critical role in the rapid immune response to respiratory viruses by leveraging cross-reactivity and their ability to mount accelerated responses. Cross-reactivity occurs when memory T cells generated in response to one pathogen recognize conserved epitopes in a different but related pathogen. This phenomenon is particularly relevant in respiratory viruses such as influenza, which frequently mutate their surface antigens but retain conserved internal proteins. Memory CD8+ T cells specific to conserved antigens can quickly recognize and eliminate infected cells, contributing to accelerated viral clearance and reduced viral load during secondary infections [23]. Studies on SARS-CoV-2 have also highlighted cross-reactivity, as pre-existing memory T cells specific to common cold coronaviruses can partially recognize SARS-CoV-2 antigens. This recognition may provide some degree of protection in individuals previously exposed to endemic coronaviruses [2]. Similarly, cross-reactive memory T cells have been shown to enhance the immune response against novel influenza strains, improving control of the infection and limiting its spread [23].

Memory T cells not only contribute to faster viral clearance, but also play a significant role in mitigating the severity of respiratory viral infections and reducing complications. For instance, lung TRM rapidly produce cytokines like interferon-gamma (IFN-γ) upon recognizing infected cells. This localized immune response helps to contain the infection early and limits tissue damage. In mouse models of influenza, memory T cells have been shown to significantly reduce disease severity and improve survival rates compared to naïve counterparts [1]. Additionally, memory T cells can protect against complications such as bacterial superinfections, which are common following respiratory viral infections. By rapidly clearing viral infections, memory T cells reduce the inflammatory environment that predisposes individuals to secondary bacterial infections [24]. For example, in RSV infections, pre-existing memory T cells can prevent severe pulmonary disease by enhancing viral clearance, though care must be taken to avoid immune overactivation, which can exacerbate pathology [6].

Influenza viruses are among the most studied respiratory pathogens regarding the role of memory T cells. Research has shown that memory CD8+ T cells targeting conserved internal viral proteins, such as nucleoproteins, provide cross-protective immunity against different strains of influenza. These cells accelerate viral clearance and reduce disease severity in heterologous infections, where the virus carries mutations in its surface antigens [23]. Mouse studies demonstrate that lung TRM play a vital role in preventing reinfection by maintaining a local immune presence in the respiratory mucosa, which is the primary site of influenza virus entry [4]. These findings emphasize the potential of T cell-based vaccines to complement current antibody-focused influenza vaccines. The SARS-CoV-2 pandemic has highlighted the importance of T cell immunity in controlling respiratory viral infections. Memory T cells specific to SARS-CoV-2 were detected in convalescent individuals and unexposed individuals, likely due to cross-reactivity with endemic human coronaviruses that cause the common cold. These cross-reactive memory T cells may contribute to milder disease outcomes in certain individuals [2]. Moreover, studies show that SARS-CoV-2-specific TRM in the respiratory tract can respond rapidly to reinfection, highlighting their protective role in controlling viral spread and limiting severe disease [24]. Other respiratory viruses, such as RSV, have also demonstrated the protective capacity of memory T cells, though with caveats. In the case of RSV, studies show that memory CD8+ T cells can clear viral infections effectively, but excessive activation of these cells may lead to immunopathology, exacerbating lung damage rather than alleviating disease [6]. This duality highlights the need to balance protective immunity and avoid overactivation of T cells in therapeutic strategies. For other coronaviruses, such as MERS-CoV, memory T cells have been detected in survivors, suggesting their role in long-term immunity, though data remain limited compared to influenza and SARS-CoV-2.

## 4. Challenges to Effective Immunity by Memory T Cells

### 4.1. Viral Evasion Mechanisms

One of the most significant challenges to effective immunity by memory T cells in influenza infections is the virus’s ability to undergo antigenic drift and shift (Figure 2). Antigenic drift refers to the gradual accumulation of mutations in the viral genome, particularly in genes encoding surface proteins such as hemagglutinin (HA) and neuraminidase (NA). These mutations can alter the epitopes recognized by immune cells, thereby reducing the efficacy of pre-existing memory T cells and antibodies [23]. For example, memory T cells that target conserved internal viral proteins may still respond, but their efficacy can be limited if mutations in other viral components alter the virus’s overall dynamics or infectivity. Antigenic shift, on the other hand, involves the reassortment of genetic material between different influenza strains, resulting in novel viral subtypes. These events can lead to pandemics, as the immune system often lacks pre-existing memory T cells capable of recognizing the new virus. The emergence of the H1N1 influenza strain in 2009 is a notable example, where a novel subtype spread globally due to the lack of pre-existing immunity in most populations [1]. Although cross-reactive memory T cells targeting conserved viral proteins can offer partial protection, the immune response is often insufficient to prevent widespread infection and disease.

Coronaviruses, including SARS-CoV-2, employ several immune evasion strategies that hinder effective immunity by memory T cells. A key mechanism is the extensive variability in the spike protein, which is the primary target for adaptive immune responses. Mutations in the receptor-binding domain (RBD) of the spike protein, as seen in SARS-CoV-2 variants of concern, can significantly reduce the effectiveness of pre-existing memory T cells and neutralizing antibodies [2]. Another evasion strategy involves the suppression of antigen presentation. Coronaviruses can downregulate major histocompatibility complex (MHC) expression in infected cells, thereby limiting the ability of memory T cells to recognize and respond to infected cells [6]. Moreover, SARS-CoV-2 has been shown to inhibit the IFN signaling pathway, which is essential for initiating robust T cell responses. By impairing innate immune activation, the virus reduces the recruitment and priming of T cells, potentially limiting the effectiveness of memory T cell responses during reinfections [24].

### 4.2. Anatomical Barriers

One major challenge to effective immunity by memory T cells is the difficulty in trafficking these cells to the respiratory mucosa, the primary site of infection for many respiratory viruses. Memory T cells, particularly TEM, rely on chemokine signals to migrate to infected tissues. However, the unique immune microenvironment of the lungs and respiratory tract, characterized by distinct chemokine and adhesion molecule expression, often impedes efficient recruitment. For example, the chemokine receptor CCR5 plays a crucial role in directing memory T cells to inflamed lung tissues during viral infections. Studies have shown that CCR5 deficiency reduces the recruitment of memory T cells to the respiratory tract, resulting in impaired viral clearance [25]. The vascular and epithelial barriers of the respiratory tract further complicate T cell trafficking. Endothelial cells in the pulmonary vasculature and epithelial cells lining the airways must express adhesion molecules like intercellular adhesion molecule 1 (ICAM-1) and vascular cell adhesion molecule 1 (VCAM-1) to facilitate T cell extravasation and tissue entry. During certain respiratory viral infections, insufficient expression of these molecules or active viral suppression of their expression may limit the influx of memory T cells, delaying immune responses and allowing viral replication to persist [4].

TRM cells are crucial for providing localized immunity in the respiratory mucosa, yet their numbers are often insufficient to mount an immediate and robust response against reinfection. In contrast to circulating memory T cells, TRM cells are retained within non-lymphoid tissues, where they establish a long-term presence at the site of previous infections. Their maintenance depends on interactions with the local tissue microenvironment, including chemokines and cytokines such as transforming growth factor beta (TGF-β), which promote their survival and retention. However, suboptimal generation of TRM cells during the initial infection or vaccine response can leave mucosal surfaces vulnerable to reinfection [4]. RSV provides a relevant example of insufficient TRM numbers. Following RSV infection, the induction of robust TRM responses in the lungs is often limited, leaving individuals susceptible to recurrent infections. This phenomenon may result from suboptimal antigen presentation, inadequate cytokine signaling, or a failure to establish the necessary cellular niches for TRM persistence [6]. Moreover, aging and chronic conditions can further diminish TRM numbers, compounding the difficulty of achieving effective immunity in older populations.

### 4.3. Waning Immunity over Time

A significant challenge to the long-term efficacy of memory T cells is the natural waning of immunity over time. This decline in the quantity and functionality of memory T cells compromises their ability to protect against reinfections with respiratory viruses. Several factors contribute to this phenomenon, including intrinsic properties of memory T cells, changes in the immune environment, and external influences such as aging and chronic infections. Over time, memory T cell populations can decrease due to limited homeostatic proliferation or apoptotic cell death. TCM cells, which rely on cytokines such as IL-7 and IL-15 for survival, may lose responsiveness to these signals, leading to a gradual reduction in their numbers. Similarly, TEM cells are subject to higher turnover rates and are more likely to diminish in the absence of periodic reactivation by antigen exposure [5]. TRM cells, which are critical for localized immunity in the respiratory mucosa, may also decline due to insufficient maintenance signals, such as TGF-β or IL-33, from the local microenvironment [4]. In addition to numerical decline, memory T cells may lose their functional capacity over time, a phenomenon often exacerbated by chronic or repeated antigen exposure. For instance, memory T cells repeatedly exposed to influenza virus antigens may undergo functional exhaustion, characterized by reduced cytokine production, diminished cytotoxic activity, and the upregulation of inhibitory receptors such as programmed cell death protein 1 (PD-1) [6]. Functional exhaustion is particularly problematic in older adults, whose memory T cells often exhibit reduced proliferative and effector capacities, leading to an increased susceptibility to respiratory viruses like RSV and SARS-CoV-2 [24]. Aging further accelerates the decline in memory T cell efficacy through a process called immunosenescence. Older individuals show reduced diversity in their TCR repertoire, limiting the ability of memory T cells to respond to new variants of respiratory viruses. Additionally, the aged immune system produces lower levels of cytokines required for memory T cell survival and homeostasis, compounding the effects of waning immunity [6].

## 5. Why Do People Still Get Sick?

### 5.1. Gaps in Memory T Cell Responses

Despite the presence of memory T cells, incomplete activation or functional exhaustion can significantly impair their ability to prevent illness during reinfections with respiratory viruses. Incomplete activation may occur if the antigen presentation process is compromised. For example, respiratory viruses like influenza and SARS-CoV-2 can downregulate MHC expression in infected cells, reducing the ability of memory T cells to recognize and respond to the virus. Without proper antigen presentation, memory T cells may fail to proliferate or produce the cytokines required to control viral replication effectively [17]. Exhaustion is another critical barrier to effective memory T cell responses. Chronic or repeated antigen exposure, such as recurring influenza or RSV infections, can lead to functional exhaustion of memory T cells. This state is characterized by reduced cytokine production, diminished cytotoxic function, and the expression of inhibitory receptors like PD-1 and cytotoxic T-lymphocyte associated protein 4 (CTLA-4). Exhausted T cells are less effective at clearing infections, which can result in prolonged illness or severe disease [6]. Exhaustion is particularly problematic in aging populations, where immunosenescence compounds the decline in memory T cell function.

The immune system’s ability to prevent illness relies on a highly coordinated interaction between memory T cells, B cells, and innate immune components. However, gaps in this coordination can undermine the protective effects of memory T cells. For instance, memory T cells depend on the support of B cells to generate neutralizing antibodies that block viral entry and limit the spread of infection. If memory B cell responses are weak or delayed, as is often the case in older individuals or during infections with rapidly mutating viruses, the overall immune response may be insufficient to prevent illness [1]. In addition, innate immunity plays a crucial role in creating the inflammatory environment needed to activate and recruit memory T cells to the site of infection. Dysregulation of innate immune responses, such as delayed IFN production, can impede the activation of memory T cells and reduce their effectiveness. For example, SARS-CoV-2 has been shown to inhibit early IFN responses, delaying the recruitment and activation of adaptive immune cells, including memory T cells [24]. A lack of coordination between these immune components can allow the virus to replicate unchecked, resulting in symptomatic illness even in individuals with prior immunity. Gaps in memory T cell responses, including incomplete activation, functional exhaustion, and insufficient coordination with B cells and innate immunity, contribute to why people continue to get sick from respiratory viruses.

### 5.2. Impact of Host Factors

The persistence of respiratory infections, even in individuals with prior exposure, can often be attributed to host-related factors such as age, comorbidities, and immunosuppression. These factors profoundly affect memory T-cell responses and the broader immune system. Aging significantly impacts T-cell immunity, reducing both naïve and memory T-cell populations, and impairing their function. In older adults, there is a marked decrease in lung-resident memory CD8+ T cells, leading to a diminished early antiviral response to influenza and other respiratory viruses [26]. Aging also correlates with an increase in immunosuppressive regulatory T cells (Tregs), which further hinder effective immune responses against RSV [27]. These age-related changes in T-cell phenotypes contribute to poorer outcomes after respiratory infections. Comorbid conditions such as diabetes, chronic obstructive pulmonary disease (COPD), and cardiovascular disease exacerbate immune dysregulation. These conditions are associated with weakened immune responses, reduced cytokine production, and diminished T-cell proliferation, all of which can impair viral clearance [28,29]. COPD patients, for instance, display reduced respiratory syncytial virus-specific CD8+ memory T-cell frequencies, which may explain their increased susceptibility to RSV infections [30]. Immunosuppression, whether disease-induced or therapy-related, also plays a critical role in undermining T-cell efficacy. In patients with HIV or undergoing treatments like chemotherapy, the number of functional memory T cells and their ability to respond to viral antigens are significantly reduced [31]. Furthermore, immunosuppressive environments delay viral clearance, increasing the risk of severe disease outcomes.

### 5.3. Emerging Infections and Lack of Pre-Existing Memory

Emerging respiratory viruses pose a significant challenge to the immune system because they often lack sufficient cross-reactivity with pre-existing memory T cells. Without prior exposure, the body fails to generate memory cells capable of recognizing and combating these novel pathogens. For example, the novel H7N9 influenza virus caused severe respiratory infections in humans due to the lack of pre-existing memory T cells, underscoring the danger of emerging influenza strains [32]. Similarly, individuals with no history of SARS-CoV or COVID-19 infections demonstrated limited T cell responses to SARS-CoV-2, further highlighting the need for novel immune responses to combat emerging infections [33]. Even when memory T cells exist, they may not provide full protection against heterologous (unrelated) viral strains. For instance, memory CD8+ T cells generated from prior influenza infections could mitigate disease severity when exposed to related strains but offered minimal protection against distinct viral subtypes [23]. The limited cross-reactivity in these cases highlights the vulnerability of populations to newly emerging pathogens with no shared epitopes. Unrelated infections can also compromise the effectiveness of pre-existing memory T cells. Studies have shown that unrelated respiratory infections, like Sendai virus, promote the apoptosis of influenza-specific lung-resident memory CD8+ T cells, reducing their ability to provide protective immunity upon subsequent exposure to influenza [34]. This effect, caused by localized inflammation and cell death, exacerbates susceptibility to reinfection and highlights the fragility of memory T cell-mediated immunity.

## 6. Enhancing Memory T Cell Responses

### 6.1. Vaccine Strategies to Boost Memory T Cells

Intranasal vaccination is a promising strategy to enhance mucosal immunity by directly targeting the respiratory tract, where many pathogens initiate infection [35]. In contrast to injectable vaccines, intranasal delivery can generate robust populations of TRM cells in the lungs. For example, an intranasal vaccine using biopolymer particles coated with influenza epitopes (BP-NP366/PA224) significantly increased lung-resident memory CD8+ T cells and improved protection against the influenza virus challenge [36]. Another recent study demonstrates that intranasal vaccination with a commercial influenza vaccine adjuvanted with NexaVant (NVT), a TLR3 agonist, effectively boosts lung CD4+ TRM cells, which are key to cross-protective, long-lasting immunity against diverse influenza strains. In contrast to traditional intramuscular vaccines, this intranasal platform enhances local mucosal immunity by promoting TRM cell formation in the lungs through a type I IFN-dependent mechanism. These CD4+ TRM cells conferred significant protection against heterosubtypic influenza infections in both mice and ferrets, highlighting the potential of this approach to overcome the strain-specific limitations of current vaccines and provide broader respiratory tract protection [37]. Another study compares lung TRM cells induced by either intranasal adenoviral vector vaccines or H1N1 influenza A infection in BALB/c mice. It finds that vaccine-induced TRM express high levels of CD103 and persist longer in the lung, while H1N1-induced TRM are shorter-lived but exhibit higher cytotoxic potential and a distinct transcriptome. Both TRM populations protect against influenza virus infections by expanding nucleoprotein-specific CD8+ T cells, and lung inflammation due to unrelated secondary virus infections do not affect the pre-existing TRM cells. The study highlights how tissue inflammation shapes TRM characteristics and offers insights for optimizing mucosal vaccines [38]. In addition, a study showed that intranasal delivery of a Modified Vaccinia Ankara (MVA) SARS-CoV-2 vaccine also induced robust pulmonary CD8+ T cell responses, essential for local viral control [39]. Another study develops an intranasal RSV vaccine using crosslinked carbon dots (CCDs) as an adjuvant and RSV prefused F protein (preF) as the antigen. The CCD/preF vaccine elicited strong serum IgG responses, mucosal immunity (including IgA antibodies, TRM cells, and antigen-specific B cells), with immunity lasting for at least one year. Combining intramuscular and intranasal immunization enhanced both systemic and mucosal immune responses, suggesting CCD/preF as a promising candidate for preventing RSV infections [40]. These studies highlight the importance of route-specific vaccine delivery to establish protective immunity at mucosal surfaces [41,42].

Vaccines designed to include conserved T cell epitopes can elicit cross-protective immunity against diverse viral strains, offering an advantage over traditional antibody-based vaccines. For instance, a CD8+ T cell-based vaccine incorporating conserved influenza epitopes demonstrated long-lasting protection against multiple influenza strains in mice, with effective viral clearance and reduced mortality [43]. Another study used recombinant live attenuated influenza vaccines to deliver RSV T cell epitopes, successfully inducing lung-localized CD8+ TRM cells that protected against both RSV and influenza viruses [44]. This approach underscores the potential of epitope-based vaccines to generate durable, broad-spectrum T cell responses. Next, a study conducted a detailed analysis of COVID-19 mRNA vaccine-induced immune memory by examining tissue samples (blood, lymphoid organs, and lungs) and found that spike (S)-reactive memory T cells were widely distributed in both lymphoid tissues and lungs, with a longer persistence in tissues compared to blood. These T cells also varied in tissue-resident marker expression depending on infection history, indicating that prior exposure shapes the localization and phenotype of memory T cells. Importantly, vaccination alone was sufficient to establish durable memory T cell populations in tissues, particularly in older individuals, where prevalence increased with age. Functionally, circulating memory T cells had stronger effector and cytotoxic profiles, while tissue-resident memory T cells adopted regulatory and localized immune profiles, helping to balance protection with minimal inflammation or tissue damage. The study highlights a functional compartmentalization of immune memory, suggesting that effective vaccination not only primes systemic immunity, but also seeds long-lived, tissue-adapted memory T cells that contribute to long-term protection [45].

### 6.2. Therapeutics Targeting T Cell Function

Cytokine-based therapies are emerging as pivotal tools to modulate T cell function, enhance memory T cell formation, and improve immune responses against respiratory viruses. Cytokines such as IL-7 and IL-15 are crucial for the survival and proliferation of memory T cells. IL-7 promotes the homeostasis of memory T cells by enhancing their metabolic activity and preventing apoptosis, while IL-15 supports the proliferation of memory CD8+ T cells without requiring antigen re-exposure [46]. These cytokines have demonstrated efficacy in restoring T cell function in chronic viral infections and enhancing antiviral immunity, making them promising candidates for respiratory virus therapeutics. Additionally, their use in combination with checkpoint inhibitors may synergistically boost T cell responses and viral clearance. Checkpoint modulation involves targeting immune checkpoint proteins such as PD-1, CTLA-4, and TIM-3 to reverse T cell exhaustion and restore antiviral immunity [47]. These inhibitory receptors are often upregulated during chronic infections and impair the functionality of virus-specific T cells. Blocking PD-1 and other checkpoints has been shown to rejuvenate exhausted T cells, leading to improved cytokine production and proliferation [48]. For example, combining PD-1 blockade with TIM-3 inhibitors enhances CD8+ T cell responses, supporting the potential of dual checkpoint therapy to combat severe respiratory infections. This strategy has also been applied in cancer immunotherapy and chronic viral infections, demonstrating the broad applicability of checkpoint inhibitors to bolster T cell-mediated immunity [47,49].

Another promising approach involves the use of IL-2 to boost memory CD8+ T cells. IL-2 plays a critical role in T cell proliferation and differentiation, and studies have shown that its administration can improve effector activity and memory formation in respiratory virus infections. A study on RSV demonstrated that IL-2 treatment significantly enhanced CD8+ T cell memory responses in the lungs, leading to better viral clearance upon reinfection. Mice treated with IL-2 exhibited reduced illness and improved antibody responses, suggesting that IL-2 could serve as an adjuvant in vaccines or immune therapies for respiratory infections [50]. Another strategy involves targeting costimulatory molecules that regulate T cell activation and survival. 41BB (CD137) is a receptor expressed on activated T cells and plays a key role in their long-term survival. Research has shown that combining a 41BB agonist with a vaccine (TriVax) led to an increase in RSV-specific memory CD8+ T cells, which persisted for over a month post-vaccination. This resulted in stronger immune protection and better recall responses during secondary infections [51]. This suggests that drugs targeting 41BB could enhance vaccine-induced memory T cell responses, making them more effective against respiratory viruses. Another promising therapeutic target is OX40 (CD134), a costimulatory receptor that enhances T cell survival and function. A study demonstrated that activating OX40 signaling led to the generation of long-lived memory CD8+ T cells in the lungs, which remained functional for over a year. These cells provided protection against lethal respiratory poxvirus infections, highlighting the potential of OX40 agonists in developing more durable immunity against respiratory pathogens [52]. Additionally, RNA interference (RNAi) technology has been explored as a tool to modulate immune responses without compromising memory T cell formation. A study found that using small interfering RNA (siRNA) to inhibit RSV replication reduced lung inflammation while still allowing for the development of strong T cell memory responses. This suggests that antiviral RNAi therapies could be used to control viral loads during acute infections without interfering with long-term immunity [53].

### 6.3. Potential Roles of Exercise and Nutrition

Exercise and nutrition play critical roles in boosting memory T cell function and enhancing immune defenses against respiratory virus infections [54]. These factors influence immune cell metabolism, cytokine production, and the ability of memory T cells to mount an effective response against pathogens. Research suggests that regular exercise and optimal nutrition can improve immune resilience by modulating both the innate and adaptive immune systems. Regular moderate exercise has been shown to enhance immune function and improve host defense mechanisms against respiratory viruses. Studies indicate that exercise can reduce the severity of influenza infections by lowering lung viral loads and enhancing respiratory host defenses. However, it has also been observed that prolonged exercise may slightly reduce CD8+ T cell memory responses, potentially due to lower viral antigen exposure during initial infection. In a study on mice, treadmill exercise reduced morbidity and mortality from influenza while attenuating lung cytokine responses and CD8+ T cell memory formation. However, despite the lower CD8+ T cell memory responses, the exercised mice still exhibited strong protection upon secondary infection [55]. This suggests that while exercise modulates immune memory formation, it does not compromise resistance to reinfection. The mechanisms underlying these effects may involve reduced inflammatory responses and improved metabolic efficiency of immune cells.

Nutrition is another essential factor influencing the immune system, with specific micronutrients and dietary components playing key roles in T cell metabolism and function. Research shows that deficiencies in key micronutrients, such as iron, can impair memory T cell functionality without affecting their generation or establishment. A study found that iron-deficient mice had significantly impaired IFN-γ production by T cells, leading to weakened antiviral responses and increased morbidity during influenza infections [56]. This suggests that maintaining adequate iron levels is crucial for memory T cells to mount effective immune responses against respiratory viruses. Additionally, a balanced diet rich in proteins, prebiotics, probiotics, and bioactive plant compounds can enhance immune responses by supporting gut microbiota and optimizing T cell metabolism. Certain vitamins and minerals, such as vitamin D, zinc, and selenium, act as regulators of immune cell signaling pathways, influencing the activation and function of memory T cells. A review highlighted how malnutrition and micronutrient deficiencies impair immune responses, making individuals more susceptible to respiratory infections and reducing the effectiveness of adaptive immunity [57].

## 7. Concerns Regarding the Safety of Memory T Cell Modulation

### 7.1. Immune Dysregulation and Cytokine Storm

Memory T cells play a crucial role in immune defense against respiratory virus infections, but their modulation raises several safety concerns. While these cells are essential for rapid immune responses upon re-exposure to pathogens, their activation and expansion can lead to immune dysregulation and lung immunopathology. In certain respiratory virus infections, such as RSV, pre-existing memory CD8+ T cells have been linked to increased lung damage and morbidity. Research has shown that RSV-specific memory T cells can trigger excessive inflammation, leading to lethal immunopathology despite enhanced viral clearance [6]. Mice that have been pre-immunized to generate strong RSV-specific memory CD8+ T cell responses suffered more severe disease upon RSV infection than those experiencing a primary infection. This counterintuitive finding suggested that instead of preventing illness, these memory cells played a role in worsening disease outcomes. In this study, the rapid clearance of RSV was accompanied by significant weight loss, lung inflammation, and in some cases, lethal pulmonary disease. This suggests that excessive immune activation, rather than the virus itself, was responsible for the increased morbidity. Further analysis revealed that the harmful effects were largely mediated by IFN-γ, a cytokine produced by activated T cells. Upon RSV infection, memory CD8+ T cells quickly released high levels of IFN-γ, triggering widespread inflammation and lung tissue damage. This excessive cytokine production resulted in an uncontrolled immune response, which ultimately proved more harmful than beneficial. To confirm the role of IFN-γ in driving disease pathology, they conducted an experiment where they neutralized IFN-γ in the respiratory tract. Remarkably, this intervention reduced morbidity and prevented mortality, demonstrating that IFN-γ overproduction was a major contributor to RSV-induced immunopathology. Interestingly, this detrimental effect of memory CD8+ T cells appeared to be unique to RSV infections. In contrast, the same RSV-specific memory T cells were protective against a normally lethal influenza virus challenge, highlighting fundamental differences in how T cell responses function against different respiratory viruses. This finding has critical implications for vaccine development, as it suggests that strategies aimed at enhancing T cell immunity may be beneficial for some viruses but could be harmful in the context of RSV [6].

Additionally, RSV has been found to impair the activation of the STAT3 pathway in memory T cells, reducing their ability to clear the infection effectively [58]. STAT3 is a critical transcription factor involved in immune regulation and plays a key role in promoting the maturation and function of memory T cells. It operates in cooperation with cytokines such as IL-10 and IL-21, which are essential for sustaining effective antiviral responses. The researchers found that RSV significantly impairs the phosphorylation of STAT3 in memory CD8+ T cells, potentially compromising their ability to mount a strong immune response upon reinfection. Infants with RSV-induced bronchiolitis exhibited lower levels of STAT3 expression in their nasal washes compared to infants infected with other respiratory viruses. This suggests that RSV actively modulates the immune system by suppressing STAT3 activation, which could contribute to the virus’s ability to persist and cause recurrent infections. Given that memory CD8+ T cells rely on STAT3-mediated signaling for their development and function, this impairment could result in weaker immune memory and reduced protection against future RSV infections. In vitro experiments further confirmed the ability of RSV to disrupt STAT3 signaling. When purified human memory CD8+ T cells were stimulated with IL-21 in the presence of RSV, their STAT3 phosphorylation was significantly diminished. This reduction in STAT3 activation had downstream effects on the immune response, including a decrease in granzyme B production. Granzyme B is an essential cytotoxic molecule that enables memory T cells to kill virus-infected cells. By reducing its production, RSV effectively weakened the cytotoxic potential of memory CD8+ T cells, making them less efficient at clearing the virus from infected tissues. The findings suggest that RSV has evolved mechanisms to subvert the host immune response by targeting memory T cell function at the molecular level. By impairing STAT3 activation, RSV prevents memory CD8+ T cells from responding robustly to reinfection, potentially explaining why RSV infections recur throughout life and why long-term immunity against the virus is difficult to achieve [58].

### 7.2. Heterologous Immunity: Pros and Cons

The modulation of memory T cells plays a significant role in shaping heterologous immunity, where immune responses to one pathogen influence the outcome of infections with unrelated viruses. This process can have both protective benefits and harmful consequences, particularly in respiratory virus infections. One of the key advantages of memory T cell modulation is its ability to provide cross-protective immunity against different respiratory viruses. Studies have shown that memory CD8+ T cells, originally generated against lymphocytic choriomeningitis virus (LCMV), can be reactivated upon vaccinia virus (VV) infection, leading to improved viral clearance and decreased mortality [23]. Similarly, heterologous immunity has been observed in influenza infections, where pre-existing memory T cells can recognize conserved viral epitopes, reducing disease severity and improving early viral control [1]. These findings suggest that modulating memory T cells could enhance vaccine strategies by broadening immunity beyond a single virus strain, making it particularly useful for combating rapidly mutating viruses such as influenza and coronaviruses.

However, the downsides of heterologous immunity modulation can be significant. One major risk is immunopathology, where an excessive or misdirected immune response leads to tissue damage rather than protection. A study found that pre-existing influenza-specific memory CD8+ T cells, when reactivated during LCMV infection, contributed to increased lung immunopathology and weight loss, despite accelerating viral clearance [59]. This suggests that while cross-reactive memory T cells can be beneficial in controlling new infections, they may also lead to unintended inflammatory responses, worsening disease outcomes. Another potential drawback is delayed or impaired recruitment of virus-specific T cells. In some cases, memory T cells specific to a prior infection can outcompete the recruitment of T cells that are actually needed for the new infection. Research on RSV found that bystander recruitment of heterologous T cells (such as LCMV-specific memory CD8+ T cells) did not enhance RSV immunity but instead delayed the recruitment of RSV-specific T cells, leading to prolonged viral persistence and increased disease severity [60]. This interference with pathogen-specific immunity raises concerns about the strategic modulation of memory T cells in vaccine design. Furthermore, heterologous immunity can lead to memory attrition, where pre-existing memory T cells are depleted or reshaped by subsequent infections. A study found that infections with unrelated viruses could quantitatively delete or qualitatively alter memory T cell populations, potentially reducing long-term immunity to previously encountered pathogens [61]. This could mean that while modulating memory T cells may enhance short-term cross-protection, it might come at the cost of weakening immunity against previously encountered viruses.

In conclusion, the modulation of memory T cells in heterologous immunity presents both opportunities and risks in the context of respiratory virus infections. On the one hand, enhancing cross-reactive T cell responses could provide broader immunity against multiple viral strains, supporting the development of more effective vaccines. On the other hand, the risks of immunopathology, interference with pathogen-specific responses, and memory attrition highlight the need for careful regulation of memory T cell activation.

### 7.3. Specific Risks with Immune Checkpoint Inhibitors

While checkpoint inhibitors can enhance immune responses, they also carry significant risks when applied to viral infections, particularly those affecting the respiratory system. One major concern is that checkpoint inhibitors could lead to uncontrolled immune activation, resulting in excessive inflammation and tissue damage rather than improved viral clearance [62]. One of the critical risks of using checkpoint inhibitors in respiratory virus infections is their potential to exacerbate immunopathology in the lungs. Studies have shown that excessive T cell activation can lead to cytokine storms, a dangerous inflammatory response characterized by high levels of pro-inflammatory cytokines such as IFN-γ and TNF-α. If checkpoint inhibitors further enhance T cell activity in such infections, they could worsen lung inflammation, leading to irreversible lung damage or respiratory failure. In addition to exacerbating lung immunopathology, checkpoint inhibitors may also increase susceptibility to secondary infections. While they do not directly suppress immune function, their ability to induce immune-related adverse effects (irAEs) often necessitates immunosuppressive treatments, such as corticosteroids, to manage excessive inflammation. This secondary immunosuppression can increase the risk of opportunistic infections such as tuberculosis, fungal infections, and bacterial pneumonia, particularly in patients with compromised lung function [47,63]. Furthermore, a study on lung cancer patients receiving checkpoint inhibitors after chemoradiotherapy found that over 30% experienced severe respiratory toxicities, including pneumonitis and tracheobronchitis, often associated with microbial infections [64].

Another major risk is the impact of checkpoint inhibitors on memory T cell differentiation and recall responses. While immune checkpoints are known to suppress T cell activity, they also play a crucial role in maintaining immune balance by preventing overactivation. Research has shown that blocking PD-L1 can lead to a restricted and oligoclonal memory T cell repertoire, meaning that fewer T cell clones dominate the immune response, reducing overall immune diversity. This narrowing of the antigen-specific T cell response may limit the body’s ability to respond effectively to new viral variants or reinfections [65]. This effect could be particularly problematic for viruses such as influenza or coronaviruses, where immune adaptation is crucial for long-term protection. Finally, checkpoint inhibitors could alter the dynamics of heterologous immunity, which occurs when memory T cells from a prior infection cross-react with a new, unrelated pathogen. While this can sometimes enhance immune protection, it can also lead to misdirected immune responses that impair viral clearance. Interestingly, a recent study examined how immune checkpoint inhibition affects cross-reactive T cell responses in chronic viral infections, specifically in mice chronically infected with LCMV and later challenged with Pichinde virus (PICV). When αPD-L1 blockade was applied to reinvigorate exhausted T cells, it did not significantly alter the cross-reactive immune response to PICV, suggesting that checkpoint inhibition may not always affect heterologous immunity [66].

### 7.4. Other Concerns

Virus-specific memory T cells can become dysfunctional over time, leading to inadequate immune responses upon secondary infections. Studies have shown that clonal expansions of memory CD8+ T cells can dominate the immune response, but these clones often have diminished recall potential, making them less effective in protecting against secondary infections [67]. Furthermore, the differentiation of memory T cells following mucosal respiratory infections is often incomplete, leading to a poor long-term maintenance of these cells in the lungs [68]. Additionally, memory T cells in the lung airways can gradually decline over time, weakening early immune responses and potentially compromising vaccine efficacy [69]. The benefits and drawbacks of boosting memory T cells are listed in Table 3.

## 8. Future Directions

The composition of memory T cell subsets plays a pivotal role in determining the effectiveness and durability of immune protection against respiratory viruses. TRM cells are of particular interest as they reside in the lungs and provide rapid, localized responses to reinfection. Research highlights the need to maintain TRM cells in respiratory tissues, as their decline compromises long-term protection against pathogens like influenza [72]. Similarly, studies emphasize the importance of cytokine-induced differentiation of effector-like memory T cells (KLRG1hi, T-bet+) for their robust pathogen control [73]. However, optimizing the balance between these subsets to ensure both breadth and durability of immune responses remains an unresolved challenge in vaccine design. Heterologous immunity, wherein memory T cells generated against one pathogen respond to a different, unrelated pathogen, is an important but underexplored phenomenon in respiratory virus infections. This concept has been demonstrated in models where prior immunity to influenza altered the immune response to LCMV, leading to both enhanced viral clearance and increased immunopathology [74]. Similarly, cross-reactive T cells have shown potential in mitigating disease severity during Zika and dengue virus co-infections, but their contributions to protection versus immunopathology in respiratory viruses like SARS-CoV-2 are still debated [75]. Unraveling how heterologous immunity shapes the host response to novel respiratory pathogens is crucial for predicting immune outcomes and designing broad-spectrum vaccines.

Single-cell sequencing has revolutionized immunology by enabling detailed profiling of T cell diversity and function at a granular level. This technology allows researchers to pair transcriptomic data with immune receptor profiling, elucidating the clonality and functional heterogeneity of T cells. Tools like scRepertoire integrate TCR data with transcriptomic datasets, offering insights into antigen-specific immune responses [76]. Another breakthrough, repertoire and gene expression by sequencing (RAGE-Seq), uses long-read sequencing to reconstruct full-length TCR and B cell receptor (BCR) sequences, providing unprecedented accuracy in identifying clonal expansions and transcriptional profiles of lymphocytes [77]. These advances offer the potential to dissect the dynamics of memory T cell formation, function, and longevity in response to respiratory virus infections. Artificial intelligence (AI) is rapidly transforming immunological research by providing tools to analyze complex datasets, model immune responses, and predict therapeutic outcomes. AI-based models, such as those integrating single-cell RNA sequencing with TCR data, can map functional T cell landscapes, identifying phenotypic shifts associated with disease or immunotherapy responses [78]. Machine learning algorithms applied to transcriptomic and immune repertoire data have been used to predict patient responses to vaccines and immunotherapies, enhancing the precision of clinical interventions [78]. These innovations not only deepen our understanding of T cell immunobiology, but also facilitate the design of next-generation vaccines and therapeutics tailored to individual immune profiles. The integration of single-cell sequencing and AI technologies holds immense promise for advancing T cell research, offering powerful tools to optimize immunological interventions and improve outcomes in the fight against respiratory viruses.

Recent advances in high-dimensional and spatially resolved technologies have transformed our understanding of the human immune response to respiratory viral infections, particularly by uncovering the complexity and localization of memory T cells in lung tissue. Spatial transcriptomics, a technology that allows for mapping of gene expression within intact tissue architecture, has been particularly impactful in this field. For example, it has enabled researchers to visualize how memory T cells are compartmentalized within the lung following viral infections like SARS-CoV-2 and influenza, revealing their proximity to epithelial structures and interaction with local immune microenvironments [79]. One case study used spatial whole-transcriptome analysis of COVID-19-infected tissues and identified persistent tissue-localized memory B and T cell responses in cancer patients, even after systemic viral clearance. This spatially anchored immunity was linked to antiviral defense as well as potential modulation of the tumor microenvironment, illustrating the dual role of tissue memory in both infection control and therapeutic response [80]. In parallel, high-dimensional single-cell and multi-omics platforms—including single-cell RNA sequencing, spatial VDJ sequencing, and integrated proteomics—are revealing the heterogeneity and functional states of lung-resident immune cells. One study integrated five spatial multi-omics technologies across COVID-19-infected lung tissues, uncovering region-specific immune responses and linking macrophage and T cell distributions with cytokine signaling patterns at high spatial resolution [81]. Similarly, spatial VDJ sequencing has allowed researchers to trace clonal expansion and lineage relationships of T and B cells within tissues, providing insight into how localized memory pools develop and function during infection and vaccination [82]. These technologies have also contributed to identifying novel markers of tissue-resident memory T cells and their distinct roles in antiviral immunity. For instance, single-cell analyses revealed that CD4+ TRM cells in nasal tissue not only persist long after influenza infection, but are also key to heterosubtypic protection and tissue preservation, with their residency governed by the CXCL16–CXCR6 axis [83].

## 9. Conclusions

While memory T cells are crucial for mitigating the severity of respiratory virus infections and enabling faster immune responses, their protective efficacy is often incomplete due to viral evasion mechanisms, waning immunity, and anatomical barriers that limit access to sites of initial infection. The recurrence of infections underscores the urgent need for more effective, durable interventions that enhance memory T cell responses—especially at mucosal surfaces where respiratory viruses commonly enter. Promising strategies, such as mucosal vaccines designed to induce robust tissue-resident memory T cell populations, offer new avenues for protection but must be rigorously validated through clinical research. Moving forward, it is essential to center research efforts around real-world patient needs and physiological contexts, ensuring that immunological insights translate into clinically viable solutions. By integrating advanced technologies with people-centered, clinically grounded approaches, we can build more effective and equitable treatments to address the persistent global burden of respiratory viral diseases.

## Figures and Tables

**Figure 1 medsci-13-00048-f001:**
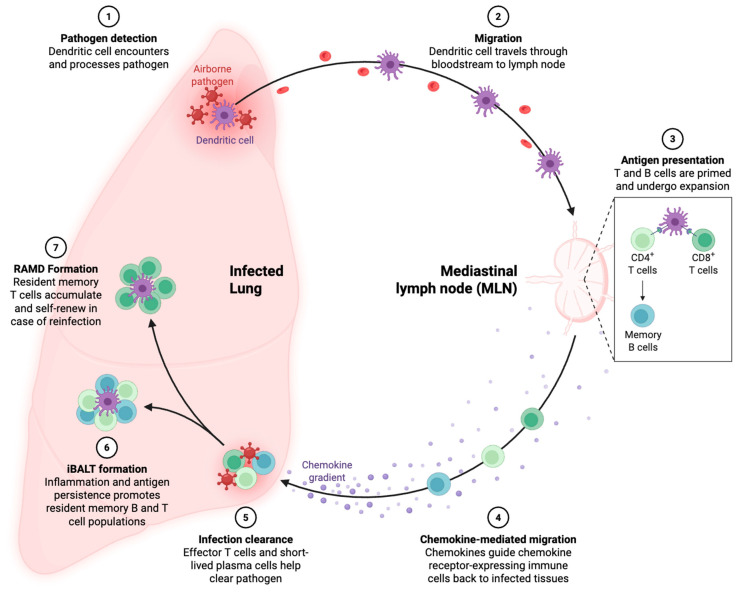
The formation and role of memory T cells in lung infection and reinfection. Initially, dendritic cells detect airborne pathogens in the lung (Step 1) and migrate to the mediastinal lymph node (MLN) via the bloodstream (Step 2). In the MLN, dendritic cells present antigens to naïve T and B cells, leading to their activation and expansion into effector and memory cell populations, including CD4+ T cells, CD8+ T cells, and memory B cells (Step 3). Chemokines guide the activated immune cells back to the infected lung, allowing them to home to the site of infection (Step 4). Effector T cells and short-lived plasma cells clear the pathogen, reducing the infection (Step 5). Persistent inflammation and antigen presence can lead to the formation of inducible bronchus-associated lymphoid tissue (iBALT), which promotes the maintenance of resident memory B and T cells (Step 6). Over time, resident airway memory T cells (RAMD) accumulate and self-renew in the lung tissue, enabling a rapid and effective immune response upon reinfection (Step 7).

**Figure 2 medsci-13-00048-f002:**
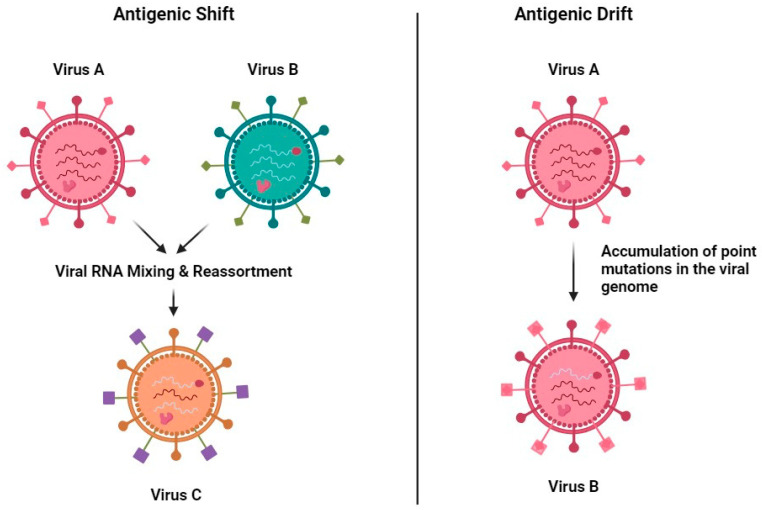
Mechanisms of viral evolution: antigenic shift and antigenic drift. On the left, “Antigenic Shift” shows the process where two different viruses (Virus A and Virus B) infect the same host cell and exchange genetic material, leading to the creation of a new hybrid virus (Virus C) with a mixed genome. This can result in a virus with a new antigenic profile, potentially leading to major outbreaks if the population has little to no immunity against it. On the right, “Antigenic Drift” describes a more gradual process where a single virus (Virus A) accumulates point mutations over time, resulting in small changes to its genome. This can gradually change the virus’s antigenic properties and may lead to seasonal flu variations as the immune system needs to continuously adapt to the new viral forms.

**Table 1 medsci-13-00048-t001:** Comparison of memory T cell types.

Feature	Central Memory T Cells (TCM)	Effector Memory T Cells (TEM)	Tissue-Resident Memory T Cells (TRM)
Location	Secondary lymphoid organs (e.g., lymph nodes, spleen)	Peripheral tissues, blood	Non-lymphoid tissues (e.g., lungs, skin, gut)
Migration Ability	High; recirculates between blood and lymphoid tissues	Intermediate; migrates between blood and tissues	Low; remains in tissue of initial infection
Cytokine Production	IL-2 (promotes proliferation)	IFN-γ, TNF-α (rapid effector function)	IFN-γ, TNF-α, and localized cytokines
Proliferative Capacity	High; capable of rapid clonal expansion upon reactivation	Moderate; less proliferative than TCM	Low; limited proliferation in situ
Longevity	Long-lived	Intermediate longevity	Long-lived but dependent on local environment
Role in Immune Response	Provides long-term surveillance; proliferates to generate effector cells upon reinfection	Rapidly provides effector functions upon antigen re-encounter	Immediate localized response to reinfection
Protection Scope	Broad, systemic immunity	Rapid protection in peripheral tissues	Site-specific immunity
Sensitivity to Antigen	Moderate; requires antigen presentation for activation	High; responds quickly to antigens	Moderate; activation influenced by tissue environment
Metabolic Profile	Relies on oxidative phosphorylation	Mix of glycolysis and oxidative phosphorylation	Primarily oxidative phosphorylation
Examples of Infections Targeted	Systemic infections (e.g., bloodborne pathogens)	Infections at peripheral sites (e.g., skin, mucosa)	Localized infections (e.g., lung influenza)

**Table 2 medsci-13-00048-t002:** Key surface markers associated with the major memory T cell subtypes.

Surface Marker	TCM	TEM	TRM
CD45RO	+	+	+
CCR7	+	–	–
CD62L (L-selectin)	+	–	–
CD69	–	–	+
CD103 (αE integrin)	–	–	+ (especially CD8+ TRM)
S1PR1	+	+	–
CXCR3	+	+	+
CX3CR1	–	+ (especially in cytotoxic TEM)	–
CD27	+	±	±
CD28	+	±	±
Integrins (e.g., VLA-1)	–	±	+
CD49a	–	–	+ (in some TRM populations)
PD-1	± (low)	± (variable)	+ (especially in barrier tissues)
IL-7Rα (CD127)	+	+	+

CD45RO: Common memory marker; CCR7 and CD62L: Guide TCM to lymph nodes; CD69/CD103: Define TRM residency in non-lymphoid tissues; S1PR1: Sphingosine-1-phosphate receptor, downregulated in TRM to prevent egress from tissues; CXCR3/CX3CR1: Chemokine receptors involved in trafficking and inflammation; CD27/CD28: Co-stimulatory molecules—often lost with terminal differentiation; CD49a: defines functionally potent cytotoxic TRM subsets, influencing both their effector functions and tissue localization; CD127: Important for homeostatic proliferation and survival across all memory subsets. + is present; – is absent; ± means can be present or absent.

**Table 3 medsci-13-00048-t003:** Pros and cons of modulating memory T cells in respiratory virus infections.

Aspect	Pros	Cons
Enhanced Immunity	Strengthens long-term protection against reinfection [70].	May lead to excessive inflammation and immunopathology, causing lung damage [6].
Rapid Response	Faster viral clearance due to quicker activation of memory T cells [71].	Overactivation of memory T cells can contribute to cytokine storms and severe disease [6].
Cross-Protection	Can provide protection against related viral strains through heterologous immunity [23].	May cause ineffective immune responses if memory T cells are not well-matched to the new virus [60].
Vaccine Enhancement	Boosts vaccine-induced immunity by generating stronger memory T cell populations [51].	Improper modulation can lead to reduced vaccine efficacy or immune exhaustion [67].
Reduction in Disease Severity	Memory T cells help reduce viral load, leading to milder infections [69].	Excessive memory T cell responses may worsen lung damage, as seen in RSV infections [6].
Long-Term Immunity	Memory T cells persist for years, providing durable immunity [52].	Over time, memory T cell responses may weaken or become dysfunctional [67].

## Data Availability

No new data were created.

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
