# Peer review of "Memory T Cells in Respiratory Virus Infections: Protective Potential and Persistent Vulnerabilities"

_medsci, 2025, doi:10.3390/medsci13020048_

Round 1
Reviewer 1 Report
Comments and Suggestions for Authors
Reviewer #1:
Respiratory viral infection represents a growing health problem. In fact, the annual number of hospital admissions due to pneumonia worldwide was estimated to be 6.8 million in 2015 and 450 million cases of pneumonia were recorded worldwide, every year, as far back as 2002. There is significant interest in understanding memory T cells, in protecting against respiratory virus infections.
Recently publications showed several research of how resident memory T cells in the lung are critical mediators of protection against respiratory viral infections.
In this manuscript, by Susanto et al titled " Memory T Cells in Respiratory Virus Infections: Protective Potential and Persistent Vulnerabilities". The authors performed an analysis of the role of memory T cells in the immune response to respiratory virus infections with the main objective of reduce the burden of respiratory virus globally.
There are several concerns that to be addressed.
This manuscript is well written and sites key findings in the field, therefore it will be helpful for investigators entering into Memory T Cells research.
The study would benefit the section on general aspects concern to Respiratory Virus Infections. Comments to improve the clarity of the manuscript are provided below.
Comments for the authors' consideration:
- Create a figure explaining the mechanisms of memory CD8 and CD4 T cells in immunity in the context respiratory virus infections.
- Concentrate in a table the main cell surface markers in memory T cells and include it in the manuscript.
- Please add in the point 6. Enhancing Memory T Cell Responses a figure explaining the therapeutic approaches in vaccines, antiviral drugs and several cytokines in the context of resolved the respiratory viral infection and inflammation.
Author Response
Respiratory viral infection represents a growing health problem. In fact, the annual number of hospital admissions due to pneumonia worldwide was estimated to be 6.8 million in 2015 and 450 million cases of pneumonia were recorded worldwide, every year, as far back as 2002. There is significant interest in understanding memory T cells, in protecting against respiratory virus infections. Recently publications showed several research of how resident memory T cells in the lung are critical mediators of protection against respiratory viral infections. In this manuscript, by Susanto et al titled " Memory T Cells in Respiratory Virus Infections: Protective Potential and Persistent Vulnerabilities". The authors performed an analysis of the role of memory T cells in the immune response to respiratory virus infections with the main objective of reduce the burden of respiratory virus globally.
We would like to express our sincere gratitude to the Reviewer for taking the time and effort to provide valuable feedback on our manuscript. We deeply appreciate the constructive comments and insightful suggestions, which have greatly contributed to improving the overall quality, clarity, and rigor of our work. Below, we provide a point-by-point response to each of the Reviewer’s comments.
There are several concerns that to be addressed.
This manuscript is well written and sites key findings in the field, therefore it will be helpful for investigators entering into Memory T Cells research.
We sincerely thank the Reviewer for the positive evaluation of our work. We are encouraged by the recognition and grateful for the thoughtful feedback.
The study would benefit the section on general aspects concern to Respiratory Virus Infections. Comments to improve the clarity of the manuscript are provided below.
Comments for the authors' consideration:
Create a figure explaining the mechanisms of memory CD8 and CD4 T cells in immunity in the context respiratory virus infections.
We appreciate the Reviewer’s comment. As shown in Figure 1, we have illustrated the formation and role of the relevant T cell subsets in the context of respiratory infection. However, if the Reviewer has specific suggestions on how the figure could be further improved or expanded, we would be grateful to receive them and would be pleased to revise the figure accordingly.
Concentrate in a table the main cell surface markers in memory T cells and include it in the manuscript.
We thank the Reviewer for this excellent suggestion. In response, we have now included a summary of the key surface markers of memory T cells in the revised Table 2 to enhance clarity and accessibility for readers.
Please add in the point 6. Enhancing Memory T Cell Responses a figure explaining the therapeutic approaches in vaccines, antiviral drugs and several cytokines in the context of resolved the respiratory viral infection and inflammation.
We thank the Reviewer for the thoughtful suggestion. However, as outlined in Section 6, our discussion focuses on approaches aimed at enhancing the function of memory T cells, rather than on the treatment of viral infections per se. Therefore, while the Reviewer’s point is valuable, it may not be directly applicable to the specific scope of this section.
Reviewer 2 Report
Comments and Suggestions for Authors
In this review, the author aims to explore the dichotomy of memory T cells in respiratory virus infections: their potential to confer robust protection and the limitations that allow for breakthrough infections. The Author highlights recent advances in vaccine strategies aimed at bolstering T cell-mediated immunity and discuss the challenges posed by viral immune evasion. This is a well-written manuscript; the author made a great effort, and I am accepting this manuscript as it is.
Author Response
In this review, the author aims to explore the dichotomy of memory T cells in respiratory virus infections: their potential to confer robust protection and the limitations that allow for breakthrough infections. The Author highlights recent advances in vaccine strategies aimed at bolstering T cell-mediated immunity and discuss the challenges posed by viral immune evasion. This is a well-written manuscript; the author made a great effort, and I am accepting this manuscript as it is.
We sincerely thank the Reviewer for taking the time to review our manuscript and for the positive evaluation of our work. We greatly appreciate the encouraging feedback.
Reviewer 3 Report
Comments and Suggestions for Authors
This review gives a clear and insightful overview of the dual functions of memory T cells in respiratory virus infection, effectively emphasizing their protective functions, such as cross-reactivity, rapid response and local immunity, as well as the remaining challenges, including virus evolution, anatomical obstacles and weakened immune response. The integration of data from recent studies on novel coronavirus, influenza and RSV contributes to the timeliness of the topic, and visual elements increase the clarity of complex immunological concepts. The discussion on therapeutic interventions, such as intranasal vaccines and cytokine-based methods, is a valuable supplement and points out the opportunities for transformation.
Suggestions for improvement:
1. Update of references: Most of the cited documents are before 2024. Although the manuscript was submitted in 2025, only a few references are from 2024. The inclusion of recent major publications, especially those in 2024 and early 2025, will enhance the relevance of the Review and ensure consistency with the latest scientific progress. Consider integrating Kwang et al.' s research (Nat Commun, 2024) and Julia et al.' s research (Immunity, 2024), and updating or replacing the old references with the latest data.
2. The paper does not point out which studies are about people and which are about animals, the discussion aspect includes organs, tissues, cells and genetic markers. The comparison between human and animal memory T cells is also a good discussion point.
3. On the transparency of graphics generated by artificial intelligence: Figure 1 seems to be generated by using artificial intelligence tools, as shown in PDF metadata. This will help to clarify this point in the graphic legend or acknowledge section, because some journals require the disclosure of artificial intelligence-assisted content creation to improve transparency.
4. Incorporate emerging technologies: By discussing new technologies to enhance our understanding of human immune response, we can strengthen the review. For example, spatial transcriptomics and other high-dimensional feature analysis methods applied to human lung tissue began to reveal new insights into the location and function of memory T cells.
5. Conclusion Revision: The conclusion can be improved, and the importance of people-centered research and the clinical verification of promising strategies (such as mucosal vaccine) can be emphasized. This will help to emphasize the way forward in the development of treatment.
6. For the sake of clarity, a concise legend is added in Figure 1 to briefly explain the various stages of memory T cell formation, which will help readers understand.
Author Response
This review gives a clear and insightful overview of the dual functions of memory T cells in respiratory virus infection, effectively emphasizing their protective functions, such as cross-reactivity, rapid response and local immunity, as well as the remaining challenges, including virus evolution, anatomical obstacles and weakened immune response. The integration of data from recent studies on novel coronavirus, influenza and RSV contributes to the timeliness of the topic, and visual elements increase the clarity of complex immunological concepts. The discussion on therapeutic interventions, such as intranasal vaccines and cytokine-based methods, is a valuable supplement and points out the opportunities for transformation.
First and foremost, we would like to extend our sincere gratitude to the esteemed Reviewer for their invaluable support and for taking the time to review our manuscript. Their insightful and constructive suggestions have greatly enhanced the quality and clarity of our work. Below, we address each of the Reviewer’s comments point by point.
Suggestions for improvement:
- Update of references: Most of the cited documents are before 2024. Although the manuscript was submitted in 2025, only a few references are from 2024. The inclusion of recent major publications, especially those in 2024 and early 2025, will enhance the relevance of the Review and ensure consistency with the latest scientific progress. Consider integrating Kwang et al.' s research (Nat Commun, 2024) and Julia et al.' s research (Immunity, 2024), and updating or replacing the old references with the latest data.
We thank the Reviewer for this helpful suggestion. The recommended references have now been incorporated and discussed in the revised version of the manuscript.
- The paper does not point out which studies are about people and which are about animals, the discussion aspect includes organs, tissues, cells and genetic markers. The comparison between human and animal memory T cells is also a good discussion point.
In this manuscript, we focus on memory T cells in humans. While we acknowledge that some studies, including the one suggested by the Reviewer, are based on animal models, we have included only those findings that are conserved across species and have demonstrated relevance to human biology. We believe that a detailed discussion on the differences between human and non-human memory T cells, although important, falls outside the scope of the current manuscript.
- On the transparency of graphics generated by artificial intelligence: Figure 1 seems to be generated by using artificial intelligence tools, as shown in PDF metadata. This will help to clarify this point in the graphic legend or acknowledge section, because some journals require the disclosure of artificial intelligence-assisted content creation to improve transparency.
We would like to clarify that Figure 1 was not generated by AI. We are unsure what led the Reviewer to this assumption. For the record, we used ChatGPT solely for grammar correction and did not rely on AI for the creation of any figures or content in the manuscript. We hope this clears up any misunderstanding.
- Incorporate emerging technologies: By discussing new technologies to enhance our understanding of human immune response, we can strengthen the review. For example, spatial transcriptomics and other high-dimensional feature analysis methods applied to human lung tissue began to reveal new insights into the location and function of memory T cells.
We sincerely thank the Reviewer for this excellent suggestion. In response, we have added the suggested topic to Section 8 of the manuscript.
- Conclusion Revision: The conclusion can be improved, and the importance of people-centered research and the clinical verification of promising strategies (such as mucosal vaccine) can be emphasized. This will help to emphasize the way forward in the development of treatment.
We appreciate the Reviewer’s comment. In response, we have rephrased the conclusion to emphasize the importance of people-centered research and the need for clinical validation of mucosal vaccines in the context of respiratory viral infections.
- For the sake of clarity, a concise legend is added in Figure 1 to briefly explain the various stages of memory T cell formation, which will help readers understand.
To enhance clarity, we have added a more detailed description of the cascade involved in memory T cell formation in Section 2.2 of the manuscript.
Round 2
Reviewer 3 Report
Comments and Suggestions for Authors
The revised manuscript continues to give a clear and insightful overview of the dual functions of memory T cells in respiratory virus infection, effectively emphasizing their protective functions, such as cross-reactivity, rapid response and local immunity, as well as the remaining challenges, including virus evolution, anatomical obstacles and weakened immune response. The integration of data from recent studies on novel coronavirus, influenza and RSV contributes to the timeliness of the topic, and visual elements increase the clarity of complex immunological concepts. The discussion on therapeutic interventions, such as intranasal vaccines and cytokine-based methods, is a valuable supplement and points out the opportunities for transformation. I appreciate the authors' thorough revisions. All of my previous concerns have been adequately addressed, and the manuscript has improved as a result. I have no further comments at this time.